# A Review of Assessment of Sow Pain During Farrowing Using Grimace Scores

**DOI:** 10.3390/ani15192915

**Published:** 2025-10-07

**Authors:** Lucy Palmer, Sabrina Lomax, Roslyn Bathgate

**Affiliations:** 1School of Life and Environmental Sciences, Faculty of Science, The University of Sydney, Sydney, NSW 2006, Australia; lpal4641@uni.sydney.edu.au (L.P.); sabrina.lomax@sydney.edu.au (S.L.); 2Sydney School of Veterinary Science, Faculty of Science, The University of Sydney, Sydney, NSW 2006, Australia

**Keywords:** pig, parturition, facial expression, welfare, dystocia

## Abstract

As the public demand for improved animal welfare in meat production systems continues to increase, better management of farm animal pain is essential. Farrowing, the process of giving birth in pigs, is known to be painful, yet the amount of pain experienced by sows that is considered “normal” and how to measure this pain is poorly understood. This review explores how pain during farrowing affects the health and welfare of both sows and their piglets, and why improved understanding and management of pain is important for animal welfare and farm productivity. A key focus was the use of facial grimace scales, a tool that can be used to assess pain by observing changes in facial expressions. The review examines the newly developed sow grimace scale that has been described as a simple and promising way to detect and score the pain experience during farrowing. Accurately identifying pain in sows could lead to improved care during birth and enhanced sow and piglet health and welfare, ultimately optimising productivity. Understanding and managing farrowing pain is not just important for ethical reasons but also benefits society by improving the productivity and sustainability of food production systems.

## 1. Introduction

Pork consumption in Australia has been consistently increasing in the past two decades, recently having overtaken beef as the second most consumed meat in the country [1]. Profitable and effective reproductive management is essential to meet the growing consumer demand for pork. Livestock producers require the majority of female animals to have offspring in order to maintain productivity. Sows are expected to have, on average, 2.4 litters per year to sustain the Australian pork industry [2]. Parturition in sows, known as farrowing, is a crucial event for producers, with significant economic implications. Furthermore, gestation and parturition are deemed high risk periods, presenting a range of health and welfare concerns for the sow and piglets [3]. Animal welfare has become an important factor to consumers as shown through increasing public concern regarding animal use in science and the treatment of livestock in farms [4]. Optimising animal welfare is therefore not only important based on an individual’s moral or ethical beliefs, but also for meeting consumer requirements due to increased consumer concerns regarding animal welfare.

The process of parturition is associated with significant pain in humans [5] and has been recognised as painful in a number of domestic animals [6]. Despite this, parturient pain in sows is highly understudied and sometimes misunderstood [7], with the current literature lacking an understanding of what level of pain is considered “normal” [8]. Pain is highly aversive, triggering a negative affective state which can be detrimental to the welfare of the animal [9]. Hence, parturient pain must be acknowledged as a significant welfare issue experienced by sows. Although a degree of pain may be inevitable during parturition, recognising its impact on the animal’s wellbeing and applying practical strategies for its management is crucial for upholding animal welfare standards [10].

The lack of validated techniques for identification and assessment of pain in animals is a major barrier to understanding the pain experience [11] and hence the implementation of treatment and management strategies to minimise the negative impact of pain on welfare. In the past decade, facial grimace scoring has been introduced as a technique of pain detection and measurement in animals based on changes in facial expressions [11]. The development of the first sow grimace scale, by Navarro et al. [12], presents a unique opportunity for future research on the pain experience of farrowing sows and its influence on reproductive outcomes. This review aims to synthesise the current literature and identify gaps in knowledge of the pain associated with parturition in sows. Additionally, grimace scoring as a technique for pain assessment will be discussed, particularly in its utilisation with farrowing sows. The reviewed literature was selected by interrogating the Science Direct, PubMed and Web of Science databases with the search criteria including grimace scale, pain, pain score, sow, farrowing, parturition, facial expression. Exclusion criteria included articles not written in English and for which the full text was not available.

## 2. Physiology of Farrowing and Parturient Pain

### 2.1. Farrowing Physiology

Parturition is considered to be a highly painful experience but is understudied in sows compared to other domestic species. The anatomical and physiological similarities between humans and pigs allow for parallels to be drawn between the experience of parturition in women and sows [7], as has been recognised by many other authors [6,7,8,13]. In women, labour is often described as one of the most painful conditions [5]. Farrowing is hence considered to cause significant pain to the sow, as supported by behavioural [14] and physiological observations [15] and the positive effects of analgesics on sow behaviour post-farrowing [16]. In women, the intensity and location of pain during labour differs significantly between individuals [17]. It could be inferred that farrowing pain would likewise be highly variable between sows, but this has not yet been explicitly studied.

The onset of farrowing is characterised by physiological and physical changes to the cervix, including cervical ripening and dilation [18]. This is followed by myometrial contractions which manually move the foetuses caudally to the cervix [18]. This first stage of farrowing is mostly associated with visceral pain, which originates from internal organs, including the uterus, and is often experienced as widespread pain which can diffuse to nearby areas including the abdominal, lumbosacral and gluteal regions [7,19]. The second stage of farrowing is marked by the onset of strong abdominal contractions which function to expel the foetuses [18]. During this stage, somatic pain becomes dominant, resulting from the distention and stretching of the pelvic diaphragm and perineum [7]. Somatic pain originates from the musculoskeletal system and external soft tissues, with the pain considered sharp and well-localised [19]. The final stage of farrowing involves the expulsion of foetal membranes, but these are sometimes expelled between piglets [20]. During this stage, the strength of contractions decreases significantly [18] so pain could be assumed to likewise decrease in amplitude, as is common in women [21]. Throughout the parturition process, visceral and somatic pain-causing stimuli are detected by nociceptive neurons, triggering the perception of pain sensations by the sow [6].

### 2.2. ”Normal” and “Abnormal” Pain During Farrowing

Compared to what is considered normal, pain has been shown to often be extremely elevated in primiparous sows [6] as evidenced by concentration of pain markers in the blood [22] and in the instance of complications and dystocia [7]. Surveys have revealed that pig farmers and veterinarians believed farrowings classified as “difficult” are highly painful, reported in one study to occur in an estimated 4% of sows and 5% of gilts in the United Kingdom [23]. In sows, dystocia is defined as difficulty during the farrowing process, occurring due to weak myometrial contractions or factors that prevent the foetus’s movement through the birth canal, often requiring obstetrical assistance [8,20]. This condition is known to cause significantly higher levels of pain compared to normal parturition [7] and can be detrimental to both sow welfare and piglet survival [3]. Dystocia in sows is highly under-researched, with the current scientific literature lacking agreement on the frequency and classification of the condition in sows and gilts. Walls et al. [8] compiled the available definitions of dystocia, all of which typically relate to farrowing duration or inter-piglet interval, highlighting the inconsistencies between studies and lack of clear guidelines for pig producers. Furthermore, this study revealed the contentious nature of dystocia occurrence, with reported prevalence ranging from 0.25 to 47% across studies and definitions [8]. The limited knowledge of dystocia in sows highlights the overall inadequate understanding of the farrowing process, particularly in regard to sow health and welfare.

### 2.3. Implications of Parturient Pain

Perinatal pain can pose several consequences to the health and welfare of sows and piglets. Pain is aversive and distressing, causing a negative affective state [9] which can be detrimental to the welfare of the sow. Parturient pain may cause sows and gilts to become restless or agitated [7] resulting in more frequent postural changes and potentially increased incidences of piglet crushing [24,25]. Furthermore, sow restlessness can decrease time spent nursing [26] and has been reported to be associated with savaging behaviour in gilts [27]. Pain can also negatively affect the physical health of sows, with implications to body condition and lactation as it is not uncommon for sow feed and water intake to be depressed postpartum [28]. This could be related to pain, which is known to decrease appetite in pigs [29]. Inadequate feed and water intake results in weight loss and decreased milk production, which negatively impacts both sows and piglets [7]. Pain during and post-farrowing is hence not only a welfare issue but also a production and animal health concern.

### 2.4. Factors Affecting Parturient Pain

Factors that are believed to affect the degree of parturient pain in sows include sow factors; parity [22] and nutrition [29], piglet factors; litter size [3], piglet size [30] and presentation [31] and parturition factors; farrowing duration [23], inter-piglet interval [32] and dystocia [7]. Many of these factors have been shown to be associated with a longer farrowing duration, but not pain specifically, so their impact on pain is based on the assumption that extended farrowing durations cause greater pain to the sow. Farrowing duration and inter-piglet intervals are considered important factors impacting the ease of parturition [33]. The average farrowing duration has been found to be approximately 130 min, but this varies significantly between sows and is heavily influenced by breed, parity and litter size [34]. A number of studies have demonstrated that prolonged farrowing duration is directly correlated with the occurrence of stillborn piglets and can negatively impact sow health [29,31]. Dystocia occurrence is directly correlated with other complications including placenta retention and postpartum dysgalactia and can cause future reproductive failure [29]. The time between the expulsion of each piglet, known as the inter-piglet interval (IPI) is likewise highly variable, but the majority of the current literature state that an interval greater than 45 to 60 min is considered abnormal and can indicate dystocia [20,32,35]. Without a validated method of measuring pain in farrowing sows, the impact of these factors on pain cannot be properly explored.

The confinement of sows in farrowing crates is a common practice but has been demonstrated to cause stress to the sow, and prevent the expression of normal maternal behaviour, including nest-building [36]. How sow housing during farrowing impacts pain severity has not been investigated, but it could be that lower oxytocin levels observed in crated sows may result in prolonged farrowing duration [37] and potentially increased pain. Confined sows are often unable to perform nest-building behaviour and have been found to have increased blood cortisol levels, a common indicator of stress [37]. The experience of stress immediately prior to farrowing and the inability to perform natural behaviours could potentially impact the pain experience during farrowing, but this cannot be determined without appropriate techniques to measure pain. Investigation into the influence of housing during the periparturient period on parturient pain is recommended in order to improve the understanding of sow welfare in farrowing crates.

### 2.5. Treatment of Parturient Pain

Pain relief options for the effective and safe management of farrowing pain are limited. Farmers report sometimes administering Azaperone, a sedative drug, and non-steroidal anti-inflammatory drugs (NSAIDs) during farrowing to minimise stress and pain [38]. Meloxicam, a common NSAID, has analgesic properties, but is not recommended to be administered pre-farrowing as it can inhibit prostaglandin production, prolonging farrowing duration [16]. Paracetamol has been demonstrated to be a safe analgesic for use in farrowing sows, but its efficacy as pain relief has not been studied [39,40]. Both meloxicam and paracetamol are transferred to the colostrum and milk so impacts of these drugs on piglets must also be considered [40]. Oxytocin is commonly administered to sows experiencing farrowing difficulties [38,41]. Whilst oxytocin is not an analgesic and therefore does not directly decrease pain, it stimulates uterine contractions, aiding in piglet expulsion and shortening farrowing duration and IPI [42]. Treatment with oxytocin therefore could be considered as a management option for farrowing pain, as the longer the farrowing duration, the more pain is experienced, particularly in cases of dystocia and breech presentation. Importantly, exogenous oxytocin administration has been associated with greater stillborn occurrence [42]. Unfortunately, there is currently no ideal analgesic for treatment of parturient pain in sows. Without a validated pain measurement technique for farrowing sows, the efficacy of analgesics cannot be accurately determined, which may be contributing to the lack of appropriate and safe pain relief available.

Despite evidence that farrowing can be an incredibly painful process, with significant impacts on production and welfare, the majority of the available literature on farrowing difficulties do not specifically address pain, highlighting a need for further research. The lack of consensus on what is considered “normal” pain during farrowing is a key obstruction to improving management and welfare of the farrowing sow. Furthermore, there is a lack of knowledge regarding whether it would be beneficial or feasible to treat or minimise “normal” or “abnormal” pain in sows, and how this could affect the outcome of parturition.

## 3. Measuring Pain in Domestic Animals

### 3.1. Pain as a Welfare Issue

Pain is a complex and undesirable sensory and affective experience which causes stress and evokes a negative mental state [9,43]. Due to the physiological similarities between vertebrate species, it is generally accepted that animals experience pain in a similar way to humans [44]. All domestic species have been confirmed to have the capacity to experience pain, as they are able to detect, react to, and respond to noxious stimuli [45]. With many animals regarded as sentient beings, pain is a significant welfare issue as it can be detrimental to an animal’s quality of life [46]. The Five Domains Model, often considered the gold standard guideline for assessing animal welfare, identifies pain as a negative experience which can be detrimental to affective state and hence welfare [47]. This model defines the welfare status of an animal as a long-term subjective state of being with biological functioning and affective components [10]. Affective state is the cumulative psychological, physiological and behavioural experience of an animal at a given time, involving emotional arousal and valence [48]. Pain is an unpleasant sensation, with negative valence, contributing to an overall negative affective state which can be observed through behavioural and physiological changes in the animal [9]. It is inevitable that animals will experience pain throughout their lives, whether that be from disease, injury, routine husbandry practices or parturition. Whilst pain can be unavoidable, it is important for the welfare of animals that it is acknowledged, and that effective and feasible methods of pain prevention or treatment are implemented to minimise the negative sensation [10]. Without accurate techniques for detecting and measuring pain, scientists, veterinarians and livestock producers are unable to ensure that an animal is not experiencing this negative affective state and hence cannot confirm the animal’s wellbeing.

### 3.2. Measuring and Quantifying Pain

In humans, pain is detected and measured via communication, most often verbal, but since animals are unable to communicate pain via language, different methods for detecting and measuring pain must be employed. Acute pain assessment in animals is generally categorised in two distinct methods; physiological and behavioural. Pain evokes a number of physiological changes via action of the hypothalamic–pituitary–adrenal axis (HPA) and the autonomic nervous system [45]. Physiological pain assessment involves measuring physiological factors associated with distress, such as heart rate or blood concentrations of stress related hormones [49]. Quantifying pain using physiological changes can be challenging as they are influenced by several other stressors independent of pain, and the process of collecting data can cause more stress to the animal [46]. Behavioural methods of pain assessment rely on the fact that animals change their behaviour in certain ways in response to painful stimuli [50]. Pain and its associated negative affective state can hence be identified and potentially quantified via observations of postural and behavioural changes. These changes associated with pain are highly species specific [51], with each species requiring its own set of guidelines regarding what behaviours may be considered normal or abnormal.

### 3.3. Facial Grimace Scales

Charles Darwin was the first to suggest that non-human animals communicated emotions such as pain via facial expressions in a similar way to humans [52]. The Facial Action Coding System (FACS) was developed in 1978 to describe the range of facial muscle movements, known as facial action units (FAUs), in humans [53]. Affective states and pain intensity were found to be communicated through facial expressions. This led to the development of grimace scales, which were first applied to laboratory mice [54] and since then a number of other domestic species [12,54,55,56,57,58,59,60,61,62,63], outlined in Table 1. Whilst pain related FAUs are species specific, there are a number which are common across species, including orbital tightening, changes in ear positioning, and tension of the facial muscles. For each FAU, grimace scales present an image for a score 0, indicating no pain, a score 1, indicating moderate pain, and a score 2, indicating severe pain. Scorers can then compare the animal’s face that they are observing to the images and descriptions on the scale. Application of grimace scoring is most well reported in rodents due to their use as human models in biomedical research, but there is increasing research on applications of grimace scales to livestock species [11].

A major limitation to knowledge about and hence treatment of pain in domestic animals has previously been a lack of validated methods of assessing pain [11]. Pigs, like other livestock species, are prey animals, having evolved to hide overt signs of pain to avoid predation [46]. This poses a major barrier to our understanding of the experience of pain in animals and highlights the need for techniques that can detect subtle indicators of pain, such as the facial grimace scale. The development of grimace scales for a range of species has provided individuals in the animal care, research and production industries unique opportunities to further the understanding of the experience of pain in animals. For example, the rat grimace scale has been used to assess the pain caused by different euthanasia methods [64] and the mouse grimace scale has been used to evaluate analgesic efficacy [65]. Unfortunately, grimace scoring is not yet routinely used outside of the research field, for example, in veterinary clinics or on farms [66]. This indicates that despite grimace scales being described as “easy to use” [46], there is still some barrier preventing the wide-scale employment of the technique. Facial grimace scoring can provide individuals with the means to accurately detect pain, which would in turn allow for improved methods for prevention and treatment of pain, but this requires further research and a push for individuals in the animal industry to implement such scales.

## 4. Measuring Pain in Farrowing Sows

### 4.1. Existing Methods of Pain Measurement During Farrowing

There has been little published research on pain measurement during farrowing. Some possible methods have been identified, including both behavioural and physiological measures. Pain indicators demonstrated relatively frequently by sows during active farrowing may include pawing, tail flicking, back arching, trembling and pulling in of the back leg [14]. These behavioural indicators require further validation to ensure they are direct responses to pain. Relying exclusively on such behaviours to detect pain is limiting as it does not allow for quantification and is difficult to validate. Vocalisations are also often clear indicators of pain in animals [67]. Sows produce distinct grunting sounds during farrowing but these vocalisations have not been found to be correlated with pain intensity or dystocia [68]. Physiological methods of measuring pain, such as blood cortisol concentration have also been explored in relation to the farrowing period in sows. Cortisol levels tend to increase during and immediately after farrowing [69], but this may be due to the generally stressful nature of parturition, rather than pain specifically [7]. This is one of the major limitations of physiological methods for pain measurement, particularly when used to study parturition, which involves many significant endocrine changes. A rise in foetal cortisol levels is the trigger of parturition in some species, and whilst it is observed in pigs, it has not been confirmed as the ultimate initiating factor, further complicating the use of cortisol to measure pain [36]. The limitations associated with the described behavioural and physiological methods of pain detection have restricted the current understanding of the farrowing process, highlighting the need for a technique which accurately quantifies the pain experience of sows.

### 4.2. Limitations in the Understanding of Pain in Sows

There are large disparities in the level of research and scientific understanding of parturition across species. When focusing on the pain associated with parturition, cows, ewes and mares typically receive more scientific attention than sows [18]. This is because parturient pain is often considered to be more significant, in terms of both production outcomes and animal welfare, in monotocous animals, such as cows and mares, compared to sows, which are polytocous [6]. The study by Mainau & Manteca [7], which aims to review parturient pain in cows and sows, predominantly focuses on cows, presenting much less information about sow parturition and pain. Dystocia is reported to occur most frequently in ewes, then mares and cows [70] hence it has received the most scientific and management-based attention in these species. In cows, dystocia is widely recognised as a major cause of postnatal complications and calf mortality [6]. Therefore, there is a larger focus on detection and management of calving dystocia and its associated pain as the condition is considered to be of greater importance in cows compared to sows. Despite the increased understanding of cow parturient pain and dystocia, there is no single widely accepted and implemented method for detecting and measuring pain during calving, with producers relying on complex behavioural and physiological factors, as is also the case for pig producers [7]. The validation and implementation of a reliable method for assessment of sow pain during farrowing could allow for the adaptation of the technique to other livestock species, where parturition dynamics are already understood to a deeper level.

### 4.3. The Sow Grimace Scale

Pain evokes a number of spontaneous reflexes in humans and animals including involuntary changes in facial expressions [53]. A range of pain-related facial actions units (FAUs) have been observed in animals, outlined in Table 1. Grimace scales are created using these facial expressions, which can then be used as a tool to detect the presence of pain and estimate its intensity. Grimace scales have been developed for piglets during tail docking and castration [60,61] and more recently for sows during farrowing [12]. The piglet grimace scale has been validated, refined and practically implemented in numerous studies [71,72,73] but nothing has been published on the sow grimace scale since its inception. Extensive literature reviews on the application of facial grimace scales to animals, such as those by Mogil et al. [11] and Evangelista et al. [74] discuss the piglet grimace scale but do not mention the sow grimace scale developed by Navarro et al. [12], which is yet to be validated by other studies.

The sow grimace scale involves five FAUs: tension above eyes, snout angle, neck tension, temporal tension and ear position, and cheek tension (Figure 1). These FAUs are ranked from 0 to 2, with 0 indicating no pain, 1 indicating moderate pain and 2 indicating severe pain. Images of sows exhibiting each FAU at each intensity level are presented on the scale, allowing individuals to recognise and grade facial expressions with ease.

In order to maintain validity for pain assessment, it is essential for each test subject to be observed without the presence of pain to establish a baseline [66]. Navarro et al. [12] used images of the sows 19 days post-farrowing as experimental controls, relying on the assumption that they were no longer in a state of pain. Images of sows faces during this time were collected and considered indicative of no pain, with a score of 0. To represent moderate pain, a score of 1, images were captured at each inter-piglet interval. To represent severe pain, a score of 2, images were selected immediately prior to each piglet expulsion. These assumptions are necessary due to the lack of a gold standard technique for measuring pain in sows, but they are supported by prior research which reported that the performance of behavioural pain indicators increased significantly during piglet expulsion [14]. These assumed pain scores were assigned by a single observer, and then eight blinded observers were given randomised facial images to rate according to the scoring system established by the initial observer to allow for the evaluation of reliability and validity [12].

### 4.4. Comparing Grimace Scales

Of the five FAUs depicted in Figure 1, all except “neck tension” have been demonstrated to be indicative of pain in the piglet grimace scale [60,61]. “Tension above eyes,” “temporal tension and ear position,” “snout angle,” and “cheek tension” have been identified as significant FAUs in a number of other species, as shown in Table 1. The majority of perinatal sows in Australia are housed in farrowing crates, providing a unique opportunity for ease in monitoring facial expressions as sows are confined in an enclosed area lying in lateral recumbency during farrowing. This has allowed the recent sow facial expression scale to introduce a new FAU, “neck tension” which has not previously been described in any species [12]. This study reported that all five FAUs had relatively high percentages of reliability and were statistically significant in relation to the moment each image was captured [12].

The “tension above eyes” FAU, which encompasses both orbital tightening and eyebrow expression, was found to be the most reliable FAU for pain assessment using the sow grimace scale [12]. This coincides with findings from grimace scoring in piglets [60] and other animals including the mouse [54], rat [55], rabbit [56], sheep [59], and ferret [62]. Whilst there is strong evidence supporting orbital tightening as a valid indicator of pain, it has been speculated that its reported superiority to other FAUs is because it is more obvious for observers to recognise [62]. It is also not possible to conclude that orbital tightening is specifically caused by pain rather than other stressful environmental or physiological factors triggering a negative affective state. For example, this FAU has been found to be associated with nausea in rats [75] and physical restraint in lambs [59]. This inability to differentiate between pain and other negative affective states is common across FAUs and species and represents a significant limitation to grimace scales [11].

The “cheek tension” FAU has been found to be statistically significant and highly reliable as an indicator of pain in both sows [12] and piglets [60]. In other grimace scales, such as the mouse [54] and ferret [62] cheek bulging was indicative of pain, but in rats [55], sheep [59], piglets [60] and sows [12], cheeks were observed to change in an opposing way, flattening when in pain, referred to as “cheek tension”. The differences in cheek FAUs across species grimace scales are shown in Table 2. This demonstrates the highly species-specific nature of pain assessment, and the need for reliable, validated grimace scales for as many species as possible.

### 4.5. Validity and Reliability of Grimace Scales

A meta-analysis on the available grimace scales for a number of domestic species to determine their validity and reliability across the current literature found that there is a significant range in the evidence of measurement properties for grimace scales across studies and species [74]. This highlights the need for further research and validation of grimace scales. Unfortunately, the lack of a gold standard method of detecting pain in animals presents a challenge to the validation of such scales [67]. For many species, including pigs, the limited number of studies on grimace scoring is a major limitation to its validation and hence adoption as a pain assessment technique [74].

Observers require substantial training and experience for the real-time detection of changes in facial expressions due to their fleeting nature [11]. For this reason, most of the current literature uses retrospective scoring, relying on images captured from video footage to perform grimace scoring. McLennan et al. [66] found that there is a difference in results between real-time and retrospective scoring, highlighting a gap in the current knowledge on the most accurate methodology for grimace scoring. Another key limitation in the literature on grimace scoring is that many published studies do not mention or describe the training undergone by individuals conducting the rating, hence reducing the reliability of the evidence for the effectiveness of grimace scoring [11].

Observer bias, which occurs when the individual conducting the pain scoring is influenced by their expectations or opinions, is a significant consideration regarding the validity of grimace scoring [76]. Assigning scores based on visual observations is inherently subjective, particularly concerning animal pain, which can trigger strong emotional reactions in some individuals. The personality, life experience, emotional intelligence and level of empathy the scorer feels towards the animal will most likely influence the score assigned [77]. This trend was demonstrated in a study which found that individuals with higher empathy towards either humans and/or animals on average assigned higher pain scores to cattle [78] and a similar study on dogs [79]. In addition to empathy, several other factors have been reported to influence the scoring of animal pain, including age [23], veterinary education and experience [80], family size [78], and emotional attachment to animals [78]. There is some evidence that observer gender impacts the perception of the intensity of pain, with women tending to assign higher pain scores than men [12,23,81]. This phenomenon has been refuted by other studies [82,83], hence requires further investigation. Artificial intelligence programs to assess facial expressions are currently being explored as a method for eliminating observer bias and hence increasing grimace scoring validity [84]. Recently, a study by Nie et al. [85] explored the use of computer monitoring and analysis of pig facial expressions to detect heat stress, highlighting the potential applications of grimace scoring beyond parturition, as well as the integration of technology to enhance such systems. Understaffing is a significant issue for Australian pork producers [86], so the capacity for grimace scoring to be automated using artificial intelligence presents a unique opportunity for the improvement of farrowing management.

### 4.6. Validation of the Sow Grimace Scale

The sow grimace scale has not yet been validated. Validation could be conducted via comparisons of pain scores with other pain measurement techniques. This could involve physiological indicators of pain, including elevated cortisol [69], C-reactive protein or haptoglobin [87], but it is important to consider that the process of collecting the necessary blood samples would cause further stress to the sow [46]. Less invasive methods of collecting physiological data, such as electroencephalogram (EEG) output has been demonstrated to change in the presence of acute pain in piglets [88] and could potentially be used to validate the sow grimace scale. Alternatively, pain scores could be validated via observations of the performance of pain related behaviours described by Ison et al. [14] or the ease of farrowing score developed by Mainau et al. [33]. What could be considered the most accurate methods of grimace scale validation is the controlled infliction of a painful stimulus, or the administration of analgesics when pain is known to be occurring [89]. As the sow grimace scale is developed for use during farrowing, controlled infliction of pain is not applicable for scale validation. Analgesics, such as NSAIDs could be administered to farrowing sows as a method of scale validation. Perceived pain should change in response to analgesics according to dose if the scores are valid [90]. This procedure could also determine if each FAU is specifically indicative of pain rather than an unrelated negative affective state. Important factors that must be considered include individual variation, farrowing dynamics and the potential impact of the drugs on piglet and sow health and farrowing dynamics.

### 4.7. On-Farm Implementation of the Sow Grimace Scale

Following further validation of the sow grimace scale, the feasibility and method of its implementation and use on-farm must be considered. The common practice of confinement of sows in farrowing crates would allow stockpersons to have a relatively unobstructed view of the face for scoring [12]. This may not be the case for all pen designs and farrowing rooms as some layouts may not allow clear vision of the face. Furthermore, many sows farrow at night, when staff may be limited or absent, meaning close observation and scoring may not be possible. The automation of grimace scoring using artificial intelligence would resolve this concern, but its implementation would be expensive. Using the grimace scale as a technique to decide if obstetric assistance is required may be the more feasible and cost-effective option for on-farm implementation. Further study is required to determine if pain score is directly related to farrowing difficulties, but if it is found to be, pain score could be used as an indication of manual assistance being necessary, in addition to conventional methods such as IPI and farrowing duration. Any on-farm use of the grimace scale would require stockpersons to be appropriately trained [11], which is another important factor to consider. Training and implementation of the sow grimace scale on farm would provide stockpersons with a relatively simple method of estimating pain presence and intensity, which would ultimately aid in the improvement of farrowing management.

## 5. Conclusions

The scientific knowledge of the pain experience and dynamics of farrowing sows is highly limited, hindering the implementation of appropriate management strategies to improve sow welfare during parturition. This review explored the current understanding of parturient pain in sows and the techniques in which it can be measured. The major barrier to improving welfare outcomes of farrowing is the lack of a reliable and easily implemented method of detecting and quantifying the severity of pain in sows. This review investigated the potential for grimace scoring to fill this gap. A sow facial grimace scale was found to be a promising tool for parturient pain assessment, but further research is required to validate the technique. It is recommended that further investigation be conducted to assess and compare farrowing pain using the sow grimace scale, specifically analysing the impact of factors such as farrowing duration, IPI, and piglet number and presentation. Assessment of the relationship between grimace score and the performance of pain-related behaviours or physiological pain measurements could be performed in order to further validate the sow grimace scale. Future studies utilising the grimace scoring method could significantly improve the understanding of farrowing dynamics and pain across stages of parturition and in instances of abnormal farrowing, potentially aiding in the development of improved management strategies and hence better production outcomes.

## Figures and Tables

**Figure 1 animals-15-02915-f001:**
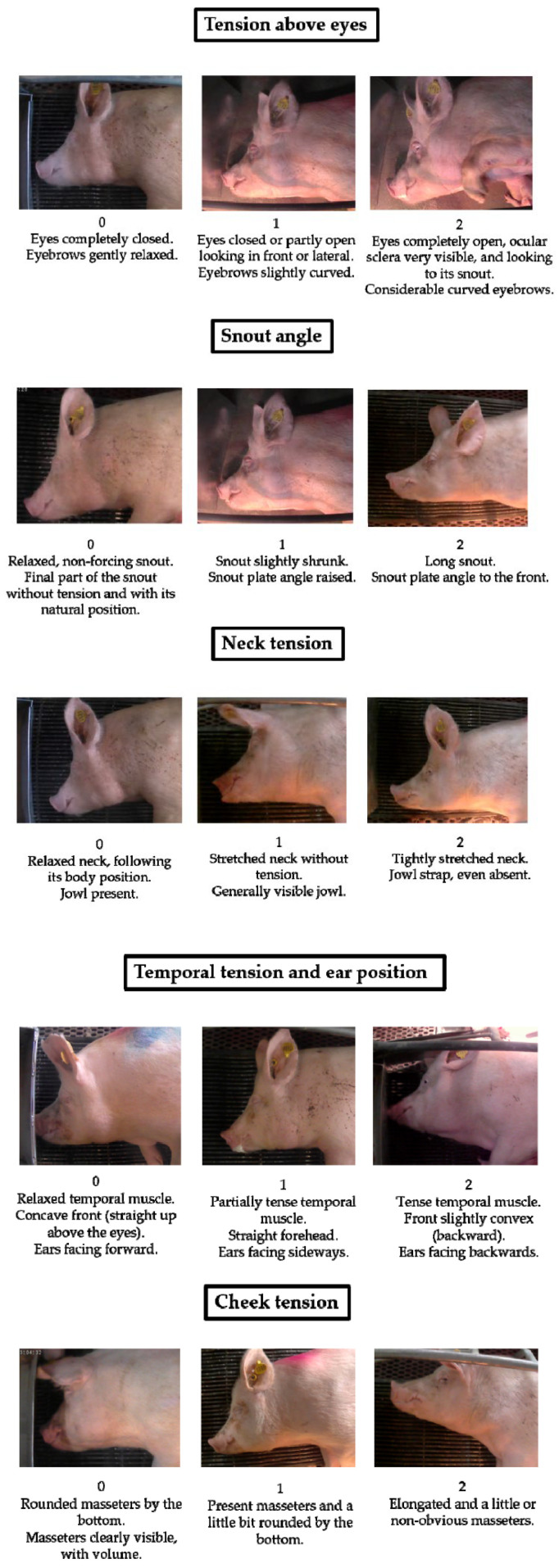
Sow grimace scale, from Navarro et al. [12].

**Table 1 animals-15-02915-t001:** Grimace scales currently published for domestic species, including the cause of pain for the animal and the Facial Action Units (FAUs) observed and scored in the study.

Animal	Cause of Pain	Facial Action Units (FAUs)
Mouse [54]	Injection of nociceptive compounds	Orbital tightening, nose bulge, cheek bulge, ear position, whisker change.
Rat [55]	Injection of nociceptive compounds	Orbital tightening, nose/cheek flattening, ear changes, whisker change.
Rabbit [56]	Ear tattooing	Orbital tightening, cheek flattening, pointed nose, whisker change.
Horse [57]	Tourniquet or topical capsaicin	Angled eye, withdrawn and tense stare, asymmetrical/low ears, square-like nostrils, muzzle tension, tension of the mimic muscles.
Cow [58]	Range of painful conditions	Orbital tightening, tense stare, tense and backwards ears, tension of facial muscles, strained or dilated nostrils, tension of the lips.
Sheep [59]	Tail docking	Orbital tightening, nose features, mouth features, cheek flattening, ear posture.
Piglet [60,61]	Castration and tail docking	Orbital tightening, cheek tightening/nose bulge, ear position.
Ferrett [62]	Intraperitoneal telemetry probe implantation	Orbital tightening, nose bulging, cheek bulging, ear changes.
Cat [63]	Admitted to veterinary hospital with abdominal pain	Whisker retraction, orbital tightening, muzzle tension, ear position, whisker position, head position.
Sow [12]	Farrowing	Tension above eyes, snout angle, neck tension, temporal tension and ear position, cheek tension.

**Table 2 animals-15-02915-t002:** Cheek Facial Action Units (FAUs) from published grimace scales with images showing the FAU in the absence of pain (pain score = 0), moderate pain (pain score = 1) and severe pain (pain score = 2). Images from [12,54,55,59,60,62].

Animal	Cheek FAU	Pain not Present	Moderate Pain	Severe Pain
Mouse [54]	“Cheek bulge”	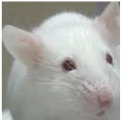	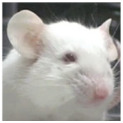	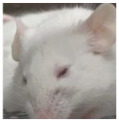
Ferret [62]	“Cheek bulging”	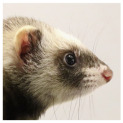	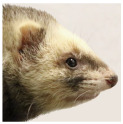	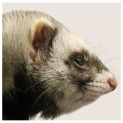
Rat [55]	“Nose/cheek flattening”	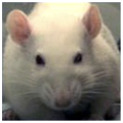	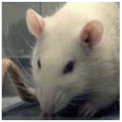	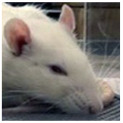
Sheep [59]	“Cheek flattening”	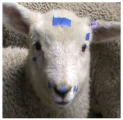	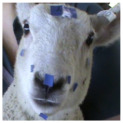	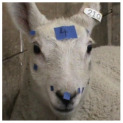
Piglet [60]	“Cheek tension”	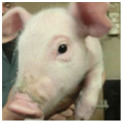	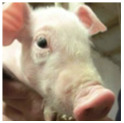	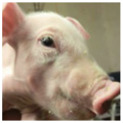
Sow [12]	“Cheek tension”	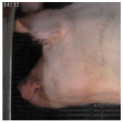	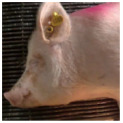	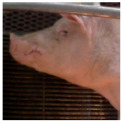

## Data Availability

No new data were created or analyzed in this study. Data sharing is not applicable to this article.

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
