# Peer review of "A Review of Assessment of Sow Pain During Farrowing Using Grimace Scores"

_animals, 2025, doi:10.3390/ani15192915_

Round 1
Reviewer 1 Report
Comments and Suggestions for Authors
Dear authors,
Please find my comments in the attached PDF document.

Author Response
Thank you for your positive comments. In response to your suggestions, we have made the below changes:
One omission is that there is no explanation of how the authors selected the literature that was used for this review. For example, inclusion/exclusion criteria, date ranges, search terms, phrasing, and article databases used. I will default to the editor as to whether this should be included in the publication.
Added the text "Additionally, grimace scoring as a technique for pain assessment will be discussed, particularly in its utilisation with farrowing sows. The reviewed literature was selected by interrogating the Science Direct, PubMed and Web of Science databases with the search criteria including grimace scale, pain, pain score, sow, farrowing, parturition, facial expression. Exclusion criteria included articles not written in English and for which the full text was not available" to clarify the source selection for the review.
There is a helpful summary of the current grimace scales for various species, although in most cases there is only one that exists for each species. As such, in terms of validation, there is no opportunity to compare which may be better (within species). There is also some variation in the different facial action units (FAUs) used to compile a scale across species. The challenge in respect of this review is that only one grimace scale for sows has been developed, so there is very little in the way of existing literature (that is directly relevant) to draw upon to write a review. Additionally, it appears the current grimace score protocol for sows has not been validated or applied widely either in research or commercially, and I think this should be emphasised more in the review. I have suggested further down some practicality issues that may be hindering the use of a grimace scoring protocol on farm.
Yes, this topic is challenging to review due to the limited existing literature, which we hope was highlighted in the review. Acknowledgement that the scale is yet to be validated is in lines 312 and 413, but a further sentence addressing this has been added to 4.3 “The piglet grimace scale has been validated, refined and practically implemented in numerous studies (Lou et al., 2022; Viscardi & Turner, 2018; Vullo et al., 2020), but nothing has been published on the sow grimace scale since its invention.”
The introduction is good, giving an overview of parturition and pain assessment techniques. What hasn’t been covered in the manuscript is what can be done about pain during farrowing, or whether any currently available methods are effective at delivering analgesia during parturition. I think it would be valuable to briefly touch on this. Are there viable options for providing pain relief during parturition? Are they safe and effective? Are there possible side effects for sows and/or piglets? It seems that if it is widely agreed that parturition is painful for sows, and there is a distinction between normal and abnormal pain (which may be identified via a grimace scale and/or other indicators) then the next step is to alleviate that pain. A brief discussion of some possibilities and/or limitations to doing so would be a valuable addition.
Added "2.5 Treatment of parturient pain
Pain relief options for the effective and safe management of farrowing pain are limited. Farmers report sometimes administering Azaperone, a sedative drug, and non-steroidal anti-inflammatories (NSAIDs) during farrowing to minimise stress and pain (Ison et al., 2016). Meloxicam, a common NSAID, has analgesic properties, but is not recommended to be administered pre-farrowing as it can inhibit prostaglandin production, prolonging farrowing duration (Mainau et al., 2012). Paracetamol has been demonstrated to be a safe analgesic for use in farrowing sows, but its efficacy as pain relief has not be studied (Kuller et al., 2021; Schoos et al., 2020). Both meloxicam and paracetamol are transferred to the colostrum and milk so impacts of these drugs on piglets must also be considered (Schoos et al., 2020). Oxytocin is commonly administered to sows experiencing farrowing difficulties (Björkman & Grahofer, 2020; Ison et al., 2016). Whilst oxytocin is not an analgesic and therefore does not directly decrease pain, it stimulates uterine contractions, aiding in piglet expulsion and shortening farrowing duration and IPI (Hill et al., 2022). Treatment with oxytocin therefore could be considered as a management option for farrowing pain, as the longer the farrowing duration, the more pain is experienced, particularly in cases of dystocia and breech presentation. Importantly, exogenous oxytocin administration has been associated with greater stillborn occurrence (Hill et al., 2022). Unfortunately, there is currently no ideal analgesic for treatment of parturient pain in sows. Without a validated pain measurement technique for farrowing sows, the efficacy of analgesics cannot be accurately determined, which may be contributing to the lack of appropriate and safe pain relief available."
Measuring pain: Being a subjective assessment, the grimace scale developed by Navarro et al. (0, 1, 2) to evaluate pain is arbitrary. Have other grimace scales used a different scoring range? Perhaps this would be an addition to Table 1.
Every grimace scale uses the same 0,1,2 scale. This information has been added in section 3.3.
If overt signs of pain are often not displayed in pigs (being a prey species), how do we reflect this in a scoring system when the ‘normal’ level of pain (e.g., during parturition) is not quantifiable?
This is a great comment and maybe the greatest challenge in this field. Levels of pain can be estimated quantitatively by factors such as blood inflammatory markers, but even this is not ideal and cannot be directly correlated to an individual's experience of pain. There is much more work to be done in this field!
Additionally, the summarised grimace scales in Table 1 cover different types of pain. As mentioned in section 2.1, there are different stages of parturition which are associated with different types of pain e.g., visceral and/or somatic. Could some FAUs be more reliable pain indicators depending on the type and intensity of pain experienced?
Another great point - there has been no work published on the differences between visceral and somatic pain measurement using FAU - probably because it is difficult to simulate these directly.
Are there ways to quantify this in combination with a grimace scale – such as with the use of EEG (electroencephalogram) or ECG (electrocardiogram) output? An expanded discussion of how grimace scores could be validated would be useful. There is little discussion critiquing the nuance of some FAU used to develop grimace scales although it does appear that observer agreement is better with some FAU compared to others.
Agreed, we have added a section about this.
4.6 Validation of the sow grimace scale
The sow grimace scale has not yet been validated. Validation could be conducted via comparisons of pain scores with other pain measurement techniques. This could involve physiological indicators of pain, including elevated cortisol [70], C-reactive protein or haptoglobin [88], but it is important to consider that the process of collecting the necessary blood samples would cause further stress to the sow [47]. Less invasive methods of collecting physiological data, such as electroencephalogram (EEG) output has been demonstrated to change in the presence of acute pain in piglets [89] and could potentially be used to validate the sow grimace scale. Alternatively, pain scores could be validated via observations of the performance of pain related behaviours described by Ison et al. [13] or the ease of farrowing score developed by Mainau et al. [33]. What could be considered the most accurate methods of grimace scale validation is the controlled infliction of a painful stimulus, or the administration of analgesics when pain is known to be occurring [90]. As the sow grimace scale is developed for use during farrowing, controlled infliction of pain is not applicable for scale validation. Analgesics, such as NSAIDs could be administered to farrowing sows as a method of scale validation. Perceived pain should change in response to analgesics according to dose if the scores are valid [91]. This procedure could also determine if each FAU is specifically indicative of pain rather than an unrelated negative affective state. Important factors that must be considered include individual variation, farrowing dynamics and the potential impact of the drugs on piglet and sow health and farrowing dynamics.
Limitations of subjective pain assessment: There are some further limitations in addition to those already noted (which included observer bias, inter-observer reliability, training). In proposing a subjective method of assessing pain based on a visual appraisal, it is possible that there is an aspect of human affinity and/or empathy or attitude towards farm animals that would influence the assessment and validity of a grimace score. Likewise, this may influence the expression of pain by individual animals depending on their experiences with humans. I think this is an important point to make. Prof Paul Hemsworth and colleagues have published on stockperson-animal interactions and how stockperson beliefs and attitudes influence their management practices, particularly in pigs.
This is a good point. We have added discussion about empathy and scoring in section 4.5.
“Assigning scores based on visual observations is inherently subjective, particularly concerning animal pain, which can trigger strong emotional reactions in some individuals. The personality, life experience, emotional intelligence and level of empathy the scorer feels towards the animal will most likely influence the score assigned. This trend was demonstrated in a study which found that individuals with higher empathy towards either humans and/or animals on average assigned higher pain scores to cattle (Norring et al., 2014).
Line 232 mentions; “Unfortunately, grimace scoring is not yet used outside of the research field, for example, in veterinary clinics or farms”. From a practicality aspect there are limitations to grimace scoring on pig farms that could explain this. Most stockpersons attending sows during farrowing are not in the habit of observing their head/face region. This would be difficult to do without entering the farrowing pen depending on the design and layout of the farrowing room. Approximately 50% of sows would be expected to farrow outside of staffed hours, thus the opportunity to assess and then potentially alleviate pain would be missed in up to half of sows. Frequent checks would be required to ensure all sows are observed given the fleeting nature of facial expressions, however this may be disruptive and stressful for sows during parturition.
We have added section which discusses this.
4.7. On-farm implementation of the sow grimace scale
Following further validation of the sow grimace scale, the feasibility and method of its implementation and use on-farm must be considered. The common practice of confinement of sows in farrowing crates would allow stockpersons to have a relatively unobstructed view of the face for scoring [12]. This may not be the case for all pen designs and farrowing rooms as some layouts may not allow clear vision of the face. Furthermore, many sows farrow at night, when staff may be limited or absent, meaning close observation and scoring may not be possible. The automation of grimace scoring using artificial intelligence would resolve this concern, but its implementation would be expensive. Using the grimace scale as a technique to decide if obstetric assistance is required may be the more feasible and cost-effective option for on-farm implementation. Further study is required to determine if pain score is directly related to farrowing difficulties, but if it is found to be, pain score could be used as an indication of manual assistance being necessary, in addition to conventional methods such as IPI and farrowing duration. Any on-farm use of the grimace scale would require stockpersons to be appropriately trained [11], which is another important factor to consider. Training and implementation of the sow grimace scale on farm would provide stockpersons with a relatively simple method of estimating pain presence and intensity, which would ultimately aid in the improvement of farrowing management.
Not only can we not quantify the level of pain that is associated with FAU, but we also cannot be sure that the affective state is indeed pain, as opposed to another mental experience. However, establishing a baseline, as was mentioned, would help identify potential changes in affective state. How could future research seek to confirm that grimace scales are identifying pain? Perhaps by administering analgesia and evaluating whether this effectively modified grimace scores?
We have added discussion about this in section 4.6.
Reviewer 2 Report
Comments and Suggestions for Authors
Dear Authors, please do some minor revisions to the work I am sending you below:
- Line 42-key words - I will add between words facial expression and word welfare, words saw pain.
- Line 46-Is the reason for the increase in pork consumption because pork production is cheaper or is it due to changes in population structure (given that some people do not eat beef for religious reasons)?
- Line 50-Is 2,4 litters per sow per year a realistic expectation? Will there be a reduction in pork imports (perhaps due to African swine fever)?
- Line 57-After words consumer requirements please add this words due to the awakening of consumer awareness regarding animal welfare.
- Line 108- Is the reason because she hasn't had any farrowing before?
- Line 197- Blood concentrations of stress related hormones-During farrowing in my opinion would cause additional stress, so I hope that nobody apply this method.
- Line 268-After word polytocous please add which represents an unjustified discrimination against pigs as a domestic animal species.
- Page 8 - there are a lot of free space on this page, so my suggestion is (if it is possible of course) to put in front of every line of FAU the name of FAU (for example in front of the first picture of Tension above eyes, you could put Tension above eyes, and to delete text Tension above eyes above the first row of pictures, etc.).
Author Response
Thank you for taking the time to review this manuscript. Please see responses to your comments, below.
Line 42 - key words. As the words 'sow pain' are already in the title of the article, we do not believe it necessary to add here.
Line 46 - this is difficult to state with a great degree of confidence, but is probably due to numerous reasons including cost, relative to beef and strong pork marketing campaigns. Regardless, this information is of interest but does not directly relate to our topic of review and so in light of keeping the article a reasonable length we suggest not adding this information.
Line 50 - you are right, many sows do not achieve 2.4 litters per year, but this is the best case scenario! Fresh pork is not currently permitted into Australia and in fact total pork imports (of processed meat) have been falling in recent years. This is probably due more to increased domestic production than external factors.
Line 57 - we have added this text. Thanks for the suggestion
Line 108 - The reasons for more pain in primiparturient mammals is not known, but to address the comment, we have added the referenced comment "as evidenced by concentration of pain markers in the blood"
Line 197 - yes, this method has been applied, using fixed venous catheters so that blood can be sampled with minimal invasion throughout farrowing.
Line 268 - this statement is quite emotive and so we have opted not to add it in.
Page 8 - thank you for this suggestion. We will adjust the formatting after we have addressed the changes from reviewers to remove excess blank space.
Reviewer 3 Report
Comments and Suggestions for Authors
Congratulations on your publication. I kindly request that the aforementioned additions and corrections be made as part of the major revision. Once implemented, they will considerably enhance the value of the paper and contribute to broadening the body of knowledge on periparturient pain in sows.
An exceptionally interesting review paper addressing the relatively rarely discussed issue of pain in farrowing sows. This subject is highly significant, particularly in the light of the growing public concern for animal welfare worldwide. The (successful) attempt to include in the publication the advances related to assessing pain severity on the basis of grimace scores deserves recognition and appreciation.
Several remarks regarding, in my view, necessary corrections and additions:
I realize that this is a difficult task; however, it would be valuable to assess the relevance of pain experienced by sows kept in different housing systems during the periparturient period. Specifically, whether and to what extent the possibility of nest building, as opposed to the lack of such an opportunity, influences the perception of pain – i.e. how this issue is manifested in farrowing pens versus farrowing crates. I would kindly request that this addition be made and that this aspect be highlighted.
If I were the authors, I would avoid direct references to pain in women during childbirth. In my opinion, this may give some readers the impression of anthropomorphism, which should be avoided.
The conclusion appears rather superficial. It would be worthwhile to clearly indicate the areas of research that ought to be undertaken, and to specify their order (according to importance), so that advancing knowledge on periparturient pain in sows becomes a realistic prospect.
I kindly request that the above-mentioned corrections be made as part of a major revision.
Author Response
Thank you for your time spent reviewing this article. We have addressed your comments as follows:
I realize that this is a difficult task; however, it would be valuable to assess the relevance of pain experienced by sows kept in different housing systems during the periparturient period. Specifically, whether and to what extent the possibility of nest building, as opposed to the lack of such an opportunity, influences the perception of pain – i.e. how this issue is manifested in farrowing pens versus farrowing crates. I would kindly request that this addition be made and that this aspect be highlighted.
Thank you for this suggestion. We have added a paragraph about this topic in the “2.4 Factors affecting parturient pain”
“The confinement of sows in farrowing crates is a common practice but has been demonstrated to cause stress to the sow, and prevent the expression of normal maternal behaviour, including nest-building (Lawrence et al., 1994). It has not yet been specifically investigated as to whether sow housing during farrowing impacts pain severity. It could be hypothesised that lower oxytocin levels observed in crated sows may result in prolonged farrowing duration (Oliviero et al., 2008) and potentially increased pain. Confined sows are often unable to perform nest-building behaviour and have been found to have increased blood cortisol levels, a common indicator of stress (Oliviero et al., 2008). The experience of stress immediately prior to farrowing and the inability to perform natural behaviours could potentially impact the pain experience during farrowing, but this cannot be determined without appropriate techniques to measure pain. Investigation into the influence of housing during the periparturient period on parturient pain is recommended in order to improve the understanding of sow welfare in farrowing crates. “
If I were the authors, I would avoid direct references to pain in women during childbirth. In my opinion, this may give some readers the impression of anthropomorphism, which should be avoided.
It is important to avoid anthropomorphism, but we think some comparison is necessary due to the lack of research about parturient pain in other mammals and inability to truly validate experience of pain in non-human animals. Comparisons to childbirth in women is also incredibly common across the literature on animal parturition. We have attempted to work the comparisons such that they are not directly relevant and have removed reference to humans from (line 109-111): Compared to what is considered normal, pain has been shown to often be extremely elevated in primiparous sows and in the instance of complications and dystocia [7], which is also seen in women [21].
The conclusion appears rather superficial. It would be worthwhile to clearly indicate the areas of research that ought to be undertaken, and to specify their order (according to importance), so that advancing knowledge on periparturient pain in sows becomes a realistic prospect.
Thank you for this feedback. We have added the following into the conclusion to address this shortfall.
It is recommended that further investigation be conducted to assess and compare farrowing pain using the sow grimace scale, specifically analysing the impact of factors such as farrowing duration, IPI, and piglet number and presentation. Assessment of the relationship between grimace score and the performance of pain-related behaviours or physiological pain measurements could be performed in order to further validate the sow grimace scale.
Round 2
Reviewer 3 Report
Comments and Suggestions for Authors
Thank you for all the changes you made to the text, especially those I requested. In my opinion, the text is now very clear, presents all the information necessary for a review of this type, and is complete. Therefore, I recommend its publication in its current form. Congratulations on choosing a very interesting, underappreciated topic, and I look forward to further publications on topics related to sow behavior during the perinatal period.
Sincerely,